# Human Adipose-Derived Stem Cell-Conditioned Medium Promotes Vascularization of Nanostructured Scaffold Transplanted into Nude Mice

**DOI:** 10.3390/nano12091521

**Published:** 2022-04-30

**Authors:** Ludovica Barone, Federica Rossi, Luigi Valdatta, Mario Cherubino, Roberto Papait, Giorgio Binelli, Nicla Romano, Giovanni Bernardini, Rosalba Gornati

**Affiliations:** 1Department of Biotechnology and Life Sciences, University of Insubria, 21100 Varese, Italy; lbarone1@uninsubria.it (L.B.); federica.rossi@uninsubria.it (F.R.); luigi.valdatta@uninsubria.it (L.V.); mario.cherubino@uninsubria.it (M.C.); roberto.papait@uninsubria.it (R.P.); giorgio.binelli@uninsubria.it (G.B.); giovanni.bernardini@uninsubria.it (G.B.); 2Department of Ecological and Biological Sciences, University of Tuscia, 01100 Viterbo, Italy; nromano@unitus.it

**Keywords:** in vivo experiment, regenerative medicine, tissue engineering, cell culture, angiogenesis, cell-free therapy, secretome

## Abstract

Several studies have been conducted on the interaction between three-dimensional scaffolds and mesenchymal stem cells for the regeneration of damaged tissues. Considering that stem cells do not survive for sufficient time to directly sustain tissue regeneration, it is essential to develop cell-free systems to be applied in regenerative medicine. In this work, by in vivo experiments, we established that a collagen-nanostructured scaffold, loaded with a culture medium conditioned with mesenchymal stem cells derived from adipose tissue (hASC-CM), exerts a synergic positive effect on angiogenesis, fundamental in tissue regeneration. To this aim, we engrafted athymic BALB-C nude mice with four different combinations: scaffold alone; scaffold with hASCs; scaffold with hASC crude protein extract; scaffold with hASC-CM. After their removal, we verified the presence of blood vessels by optical microscopy and confirmed the vascularization evaluating, by real-time PCR, several vascular growth factors: CD31, CD34, CD105, ANGPT1, ANGPT2, and CDH5. Our results showed that blood vessels were absent in the scaffold grafted alone, while all the other systems appeared vascularized, a finding supported by the over-expression of CD31 and CDH5 mRNA. In conclusion, our data sustain the capability of hASC-CM to be used as a therapeutic cell-free approach for damaged tissue regeneration.

## 1. Introduction

Three-dimensional bio-scaffolds can play a critical role in the development of new tissue by actively interacting with stem cells. Indeed, it was initially speculated that mesenchymal stem cells (MSCs) administered together with a biocompatible scaffold would engraft and differentiate into functional cells, resulting in the regeneration of damaged tissues [1]. In this regard, abounding investigations on biomaterials-MSC systems have been conducted and lot of new data have been generated, especially with regard to the in vivo effects that 3D scaffolds could have on MSCs [2,3,4].

Unfortunately, despite the FDA approval of the routine use of hematopoietic stem cells to treat patients with cancer and disorders of the blood and immune system, such as leukemia and lymphomas [5], stem cell therapies remain constrained by significant scientific and technical hurdles that hinder their expansion and limit the number of patients who could benefit from their use [6]. In fact, contrarily to expected, stem cells, when used for the regeneration of damaged tissues, do not engraft in significant numbers or for enough time to directly sustain tissue replacement [7]. In addition, only the cells within a distance of 100–200 μm to the nearest capillaries receive sufficient oxygen and nutrients to maintain their metabolism and function [8], but an optimal vascularization is necessary to promote tissue regeneration of the damaged area [9,10]. Another aspect to consider is that some authors have demonstrated, both in vitro and in vivo, that MSCs can exert an anti-angiogenic function in the wound-healing process. The capillary degeneration seems to be due to apoptosis of endothelial cells caused by reactive oxygen species (ROS) generated by MSCs; this process, however, requires a direct contact between MSCs and endothelial cells and only minimally occurs through paracrine pathways involving soluble factors such as IL-6, IL-10, TGF-β1, and TSP-1 [11,12]. Considering all the above, it is clear that it is urgent to focus the research on cell-free therapy and on the development of powerful cell-derivative systems to use in regenerative medicine [13].

Even though the past decade has been dedicated to the mechanisms through which cultured MSCs promote tissue repair, particular attention has been paid to the paracrine activity of secreted proteins and to the role of exosomes or microvesicles in this process [14,15,16]. 

These circumstances drove the research efforts in the direction of using angiogenic growth factors, such as VEGFA, IL6 and TGF-β1, adhesive peptides, extracellular matrix proteins, cytokines, or hormones combined with bio-scaffolds to promote new vessel formation [17,18].

In adulthood, blood vessels are quiescent, and angiogenesis is reactivated, in a highly orchestrated manner, within a time window of eight days following wounding [19]; in this delicate scenario, the cooperation of the extracellular matrix (ECM) with cell surface receptors is fundamental [20,21]. It is well known that the ECM is characterized by a molecular architecture with peculiar physical–chemical characteristics that confer elasticity and stiffness, whose mechanical signals affect fundamental cellular processes, including proliferation, migration, and differentiation [18,22,23]. In this scenario, the use of stem cells to promote angiogenesis and tissue regeneration could be a source of further uncontrollable events, and this supports, once again, the need to focus on the use of cellular derivatives.

In this work, we would like to demonstrate that a collagen nanostructured scaffold, loaded with a culture medium conditioned with Adipose Stem Cells (hASCs), has the same effectiveness as stem cells in supporting angiogenesis. hASCs are a subpopulation of MSCs that are easily isolated from adult adipose tissue, which is accessible in large quantities with minimal invasive harvesting procedures, and efficiently expanded in vitro [7,24].

More specifically, in this study, we report the results of in vivo experiments on athymic BALB/c nude mice grafted with a collagen-glycosaminoglycan biodegradable matrix (Integra^®^ Flowable Wound Matrix) in four different conditions. The data obtained demonstrated that Integra^®^FWM associated with hASC-conditioned medium showed the same efficiency of Integra^®^FWM associated with hASCs in promoting cellular invasion and capillary growth.

## 2. Materials and Methods

Integra^®^ Flowable Wound Matrix (FWM), kindly provided and characterized by LifeSciences Corporation (Plainsboro, NJ, USA, https://www.integralife.com), is a 3-D porous biocompatible matrix comprised of granulated cross-linked bovine tendon collagen and glycosaminoglycan. Quantitative measurements revealed that the fibers exhibited a 2 μm thickness and a length of 120 μm with an average interfiber distance of 45 μm [25]. It is commonly used for the treatment of tunneled and irregular wounds, which are often associated with excessive scar tissue formation. The device, supplied freeze-dried, can be hydrated with culture media. In this way, the scaffold acquires a gel-like consistency, making it optimal for subcutaneous injection. More information about the structure, porosity, and biocompatibility is available on the Integra website [26]. 

### 2.1. Animals

Seven-week-old athymic BALB-C nude mice (Crl:CD1-Foxn1nu086) were obtained from Charles River (Calco, Lecco, Italy). The animal studies were approved by the University of Insubria Ethical Committee (OPBA) and by the Italian Ministry of Health in accordance with the Italian D.Lgs 26/2014. In total, twelve animals were used for these experimental purposes.

### 2.2. hASC Isolation and Culture

hASCs were obtained from mammary adipose tissue of two healthy women (40 and 57 years old) who underwent breast reduction surgery. The subjects gave their informed consent for inclusion in the study. They were nonsmokers, had no history of metabolic disorders, were not taking medications at the time of the medical procedure, and had not experienced any great weight loss from dieting (Body Mass Index was <20 kg/m^2^).

hASCs were isolated according to Gronthos and Zannettino’s protocol modified in our laboratory [27]. Briefly, the stromal vascular fraction (SVF) was obtained by collagenase type II digestion (Sigma Aldrich, Milano, Italy) at 37 °C for 1 h in agitation. SVF was filtered (100 μm cell strainers) and centrifuged at 180× *g* for 10 min (Eppendorf 5804R, Milan, Italy); the resulting pellet was washed with erythrocyte lysis buffer (154 mM NH4Cl, 10 mM KHCO3, and 1 mM EDTA), then seeded in T25 flasks maintained at 37 °C, 5% CO_2_. After 6 h, nonattached cells were removed. Cells were grown in DMEM:DMEM F12 1:1 (Sigma Aldrich, Milano, Italy), supplemented with 2 mM L-Gln, 1% penicillin-streptomycin, 0.1% gentamicin, and 10% FBS. Due to the abundant number of hASCs necessary to run all the experiments and taking into account that P3 to P5 are considered as early passages expressing the same secretome [28], cells were subsequently cultured in T75 flasks and used at P5 for all experiments.

### 2.3. hASC Characterization

Cells were characterized by fluorimetric analysis (FACS), immunostaining, and quantitative polymerase chain reaction (qPCR); the protocols were described in Cherubino et al. 2016 [7] and Borgese et al. 2020 [18].

Briefly, the FACS analysis was conducted using a series of monoclonal antibodies specific for staminal markers CD44, CD90, and CD105, for the differentiation marker CD45, and for major histocompatibility molecules including HLA class I (HLA-A,B,C) and class II (HLA-DR). For immunostaining, CD44 antibody was used as a stemness marker and adiponectin receptor 1 (ADIPOR1) antibody as an adipogenic differentiation marker. 

For qPCR, CD44 and CD90 genes were used as positive stemness markers, while fatty acid-binding protein 4 (FABP4), adipocyte complement-related protein 30 (ACRP30), and acetyl-coenzyme A synthetase 2 (ACSS2) genes were examined as differentiation markers. According to the method of Palombella et al. 2017 [29], glyceraldehyde-3-phosphate dehydrogenase-GAPDH and beta2-microglobulin-β2m were used as reference genes. Quantification was conducted by using the 2^−ΔΔCt^ method. Each experiment was repeated three times.

### 2.4. hASC-Conditioned Medium and Protein Extract Preparation

hASC-conditioned medium (hASC-CM) was prepared as described in Marcozzi et al. 2020 [14]. Briefly, when hASCs reached 70–80% confluence, the medium was removed, and cells were washed twice with PBS. FBS-free DMEM was added, and cells were incubated for 48 h. The medium (hASC-CM) was removed, centrifuged at 2000× *g* for 10 min to avoid contamination of cell fragments, and then stored at −80 °C until use. 

The hASC-derived protein extract was prepared as follows: when hASCs reached confluence, cells were harvested and centrifuged at 1000× *g*, the pellet was suspended in 3 mL of fresh serum-free medium, and cells were mechanically lysed. The protein extract was used immediately after preparation. 

### 2.5. hASC-Conditioned Medium and Protein Extract Characterization

hASC-CM, derived from two different subjects, was characterized by Enzyme-Linked Immunosorbent Assay (ELISA). The assay, performed following the manufacturer’s instructions (FineTest^®^, Wuhan, China), was conducted on VEGF-A (Vascular Endothelial Growth Factor A), IL-6 (Interleukin-6), HIF-1α (Hypoxia Inducible Factor-1α), and TGF-β1 (Transforming Growth Factor-β1). The protein amount was determined recording the absorbance at 450 nm using the GloMax^®^ Discover Microplate Reader (Promega, Milano, Italy). Each experiment was repeated 3 times and the values, expressed as ng/culture medium deriving from 1 × 10^6^ cells, were reported as mean ± S.D.

An appropriate volume of the hASC-derived protein extract, prepared as indicated in Section 2.4, was assessed by ELISA to evaluate the amount of VEGF-A, IL-6, HIF-1α, and TGF-β1. Each experiment was repeated 3 times and the values, expressed as ng/1 × 10^6^ cells, were reported as mean ± S.D.

### 2.6. Xenogenic Grafting

The grafting was carried out on mice, anesthetized by isoflurane inhalation, by making 0.5 cm incisions in the backside, between the muscle and subcutaneous layer, and using a syringe with a luer-lock connector and a flexible injector. The grafts consisted of: (1) Integra^®^ FWM hydrated with fresh culture medium depleted of FBS; (2) Integra^®^ FWM hydrated with fresh culture medium containing 3 × 10^6^ hASCs; (3) Integra^®^ FWM hydrated with fresh culture medium containing crude protein extract derived from 3 × 10^6^ of hASC; (4) Integra^®^ FWM hydrated with hASC-CM derived from 3 × 10^6^ cells. Each syringe contained a total volume of 3 mL of each formulation, and a volume of 200 μL, for each preparation, was injected in any single animal. Incisions were then stitched using surgical sterile strips (Figure 1A).

### 2.7. Gross Examination of Scaffold

After 28 days from the grafting, mice were sacrificed in a CO_2_ chamber. The scaffolds were harvested, observed by a circular lens (Canon EOS 550 D), and images captured (Figure 1B). 

### 2.8. Sample Collection

For optical observations, a small piece of each sample was fixed in 1% Karnowsky solution in 0.1 M sodium cacodylate buffer (pH 7.4) at 4 °C and preserved in the same buffer. 

For molecular analysis, the samples were stored at −80 °C until RNA extraction.

### 2.9. Optical Microscopy 

In order to evaluate the formation of new vessels in the inner portion of the scaffold, fixed samples, dehydrated with ethanol (70, 90, 95, 100%), were embedded in paraffin and cut using an RMC-RM3 rotary microtome (TiEsseLab, Milan, Italy). Five nonconsecutive sections (5 μm) per sample were mounted on glass slides, stained with H&E solution, following classical procedures, and analyzed by counting the number of capillaries present in three-microscope fields of each sample. Vessels were numbered and classified as large (d > 100 µm), medium (100 > d > 20 µm), and small (d < 20 µm) capillaries using ISCapture software. 

### 2.10. RNA Extraction, Reverse Transcription, and Real-Time PCR

Total RNA was manually isolated by Trizol extraction and precipitation. The RNA was quantified by the QuantiFluor^®^ RNA System (Promega, Milano, Italy) and its integrity was assessed by 1% gel electrophoresis. The RNA was then reverse-transcribed using the iScript™ cDNA Synthesis Kit (BioRad, Milano, Italy) and the cDNA was stored at −20 °C until use. qPCR was conducted using iTaq Universal SYBR^®^ Green Supermix (BioRad, Milano, Italy). The analysis was performed on genes involved in angiogenesis such as Cluster of Differentiation 31 (CD31), Cluster of Differentiation 34 (CD34), Cluster of Differentiation 105 (CD105), Angiopoietin 1 (ANGPT1), Angiopoietin 2 (ANGPT2), and Cadherin 5 (CDH5); furthermore, the expression of the human stem marker Cluster of Differentiation 90 (CD90) and differentiation marker Fatty acid binding protein-4 (FABP4) was also evaluated. The Beacon Designer Program (BioRad, Milano, Italy) was used to design the primers used in this work and whose sequences are shown in Table 1. 

Each sample was prepared as reported in Rossi et al. [30]. Briefly, 1 μL (5 ng) of cDNA, 1 μL of forward and reverse primer mix (6 μM), 7.5 μL of SYBR Green Supermix (2×), and water to a final volume of 15 μL were mixed and placed in the CFX 96 Thermocycler (BioRad, Milano, Italy).

Values were normalized with two reference genes, Glyceraldehyde Phosphate Dehydrogenase (GAPDH) and β-actin, according to the method of Palombella et al. [29] and quantified by using the 2^−ΔΔCt^ method. Each experiment was repeated three times.

### 2.11. Statistical Analysis

Vessel counting and classification were reported as mean of vessel/mm^2^ ± standard error.

qPCR data analysis was performed by Student’s *t*-test on the ΔCt values of the scaffold supplied with the formulation (see Section 2.6 Xenogenic grafting) versus scaffold suspended in fresh culture medium, used as control. Data were expressed as mean values (±standard error) and were considered significantly different at *p* < 0.05.

## 3. Results

### 3.1. hASCs and Conditioned Medium Characterization

The hASCs used in this study were previously characterized by Cherubino et al. 2016 [7] and Borgese et al. 2020 [18].

The characterization, in terms of most known growth factors, of conditioned medium (CM) and protein cell extract, conducted by ELISA, is displayed in Figure 2. The data showed that the amount of HIF-1α was the most abundant both in CM (panel A) and in protein cell extract (panel B). VEGF-A, IL-6, and TGFβ1, although present in smaller quantities, were still represented in CM. 

### 3.2. Scaffold Evaluation and Gross Examination

For the scaffold analysis, we considered FWM as the negative control (no vascularization is expected), and FWM loaded with hASCs was the positive control (vascularization is expected). With these specifications, all the evaluations were conducted by comparing the four preparations with each other.

Upon macroscopic inspection, conducted 28 days after the grafting, all the samples appeared yellow, soft on palpation, and of comparable size. As shown in Figure 3, it should be noted that clear differences were observed in term of vascularization. In the specimen constituted by FWM alone, the blood vessels were scarce and remained confined to the periphery of the scaffold (Figure 3A). Conversely, in FWM loaded with hASCs, or hASC protein extract or hASC-CM, the vascularization was more evident and penetrated inside the scaffold (Figure 3B–D). It should be noted that, in these last three conditions, the blood vessels developed in a centripetal mode, branched within the scaffolds, and were functional. 

### 3.3. Optical Microscopy 

The optical microscopy analysis was conducted for each sample on five nonconsecutive sections to evaluate the formation of newly formed vessels in the scaffolds. Some representative microphotographs, referring to internal areas of the four different systems, are reported in Figure 4. The results of FWM alone, illustrated in Figure 4A, showed the presence of fibroblasts, collagen fibers, and the absence of vessels. In Figure 4B–D, the outcomes of FWM combined with hASCs, cell protein extract, or cell-conditioned medium are presented. In all the panels, in addition to collagen fibers and cell nuclei (probably of fibroblasts), the presence of numerous capillaries full of erythrocytes was notable. 

Vessels were then numbered (n°/mm^2^) and classified as large (d > 100 µm), medium (20 < d < 100 µm), and small (d < 20 µm). The results, shown in Figure 5, demonstrated that, although some discrepancies were appreciated among the three formulations, no statistically significant differences were observed in the number of capillaries, which stood at about 20 vessel/mm^2^, as well as in their classification based on size. In our experimental conditions, we did not find capillaries larger than 100 µm.

### 3.4. Gene Expression

The relative gene expressions of CD31, CD34, CD105, ANGPT1, ANGPT2, and CDH5 are reported in Figure 6. The most relevant results refer to the mRNA expression of CD31 and CDH5 that appeared higher in the scaffold loaded with conditioned medium (black bar) and in the one with cell protein extract (dark grey bar) compared to FWM alone. No significant differences were observed for CD34, CD105, and ANGPT1 and 2 (*p* < 0.05).

mRNA evaluation of human CD90 and FABP4 appeared undetectable in our experimental conditions. 

## 4. Discussion

Tissue engineering is a promising field that uses scaffolds to regenerate damaged tissues [31]. A good scaffold should be biocompatible, degradable, possess good porosity for the transport of nutrients and wastes, and have favorable affinity with the target tissue [32,33,34].

This, together with the knowledge that biomaterials can affect fundamental cellular processes [22,23], directed researchers to focus their attention on the use of bio-scaffolds in combination with cellular derivatives rather than MSCs [17,18].

Indeed, it is well known that during the healing mechanism, Platelet-Rich Plasma (PRP) provides a fundamental contribution to neovascularization, and some preclinical and clinical studies support the use of this cell-free method to stimulate and accelerate the angiogenic process [35,36]. Furthermore, despite some controversies regarding the PRP preparation, recent clinical trial results have demonstrated promising clinical benefits [37]. A further step toward tissue regeneration consists in the use of the secretome. Indeed, the use of a medium enriched with the secretion of MSCs has been considered as “the new paradigm towards cell-free therapeutic mode for regenerative medicine” [38]. 

To our knowledge, most of the recent literature, although abundant, consists of review articles reporting strategy hypotheses to obtain vascularization of the implanted scaffolds [9,37,39,40,41]. Vice versa, only a few papers are reporting in vivo experiments on angiogenic processes using bioactive molecules combined with scaffolds [7,19,42,43]. 

In this context, we report here the results of in vivo experiments, using hASC-conditioned medium (hASC-CM), which consists of an assortment of proteins and extracellular vesicles secreted by a cell population into the extracellular space and considered as an extremely valuable therapy in tissue regeneration [44,45]. The experiments were conducted on athymic BALB/c nude mice grafted with a collagen-glycosaminoglycan biodegradable matrix (Integra^®^FWM) associated, or not, with hASC-CM, in which VEGF-A, HIF-1α, IL-6, and TGF-β1, biomolecules with an important role in the angiogenetic process, are well represented. More precisely, as reported by De Pascale [36] and Gnecchi [46], VEGF-A stimulates the proliferation and migration of endothelial cells to form immature vasculature, whilst TGF-β1, a pleiotropic cytokine that regulates different cellular functions including proliferation, differentiation, and migration, is involved in vessel maturation [47]. Furthermore, Ramakrishnan et al. [48] reported that IL-6 induction up-regulates the expression of VEGF-A, and HIF-1α up-regulates both VEGF-A and bFGF (basic Fibroblast Growth Factor), a multifunctional protein that regulates endothelial-promoting angiogenesis and the formation of new blood vessels from the pre-existing vasculature. Therefore, we can consider hASC-CM a good formulation to promote tissue vascularization.

Gross examination and optical microscopy observation of the scaffolds, conducted 28 days after the grafting, indicated that Integra^®^FWM alone did not present internal vascularization. As the collagen present in the scaffold in conjunction with the endogenous VEGF participates in coordinate vessel sprouting [49], the absence of vascularization indicates that the amount of endogenous VEGF is insufficient to promote vessel formation inside the scaffold. Conversely, the scaffolds that were loaded with hASCs, hASC protein extract, or hASC-CM appeared highly vascularized.

Quantitative PCR analysis indicated that the expression of CD31 and CDH5 mRNA increased in the scaffolds combined with cell protein extract, but even more with their derivatives. A better performance of these two preparations versus cells can be explained by the fact that stem cells do not engraft in significant numbers or for sufficient duration to directly sustain angiogenesis [7]. CDH5, also known as vascular endothelial cadherin, is indispensable for proper vascular development and for the maintenance and control of endothelial cell contacts [50]. Furthermore, the interaction of CDH5 with VEGF is essential for the endothelial survival [51]. CD31, also known as the platelet endothelial cell adhesion molecule, is highly expressed on the surface of mature endothelial cells and is thus a well-established marker for the monitoring of vessel density in tissues [47,52]. Taken together, these data support the hypothesis that the capillaries observed in the loaded scaffolds were mature and functional.

By contrast, our results did not evidence notable variations for what concerns the other markers that we investigated. In particular, CD34, a common marker for identifying human hematopoietic stem cells, is also expressed in tip cells. Tip cells grow out from a pre-existing vascular network and are the leading cells during sprouting angiogenesis to form a new vessel [53,54]. Presumably, CD34 expression did not increase, because the capillaries observed in the loaded scaffolds were mature. CD105 (Endoglin), a cell membrane glycoprotein, is an accessory receptor for TGF-β1 [46,55], and according to the Mesenchymal and Tissue Stem Cell Committee of the International Society for Cellular Therapy, human MSCs must express CD105 to be defined as such [56]. Nevertheless, CD105 expression is upregulated in actively proliferating endothelial cells, and therefore, CD105 has been suggested as a powerful marker of neovascularization [57] and used to evaluate vessels density [58]. On these bases, CD105 can also be considered an indicator of early stages of vascularization. Then, similarly to CD34, CD105 expression levels did not increase, because, 28 days after the grafting, the vessels were mature and operating. Similarly to CDH5, ANGPT1, by the phosphorylation of the tyrosine kinases receptor Tie2, is involved in vascular remodeling and protection through the tightening of endothelial cell junctions [59]. Conversely, ANGPT2 inhibiting the ANGPT1-induced Tie2 phosphorylation, leads to the disruption of blood vessels [60]. This being the case, the reduced mRNA expression of ANGPT2, found in the preparation FWM-hASC-CM, supports the idea that this preparation could be favorable in promoting angiogenesis.

Moreover, in our previous paper [18], we demonstrated that, in the presence of the scaffold, cell proliferation and the exocytosis of factors involved in the angiogenesis process are reduced in favor of the expression of those genes involved in hASC differentiation. Considering that this paracrine event is the primary mechanism exerting the beneficial effects on injured tissues, replacement of hASCs with its CM may be convenient. It has also been reported [6,7] that the transplanted MSCs do not engraft in significant numbers or for sufficient duration to directly sustain tissue replacement. The results obtained in our experimental conditions sustain the foregoing considerations. More precisely, we found that the mRNA expression of CD90 and FABP4 appeared undetectable in the removed scaffolds. As CD90 is a marker to evaluate the presence of hASCs and FABP4 is used as a marker for differentiated adipocytes, their absence means that after 28 days from the grafting, the transplanted cells are no longer present either as stem cells or as differentiated cells. All together, these results confirm that the presence of stem cells is not necessary to promote the process of angiogenesis and sustain the capability of the combination scaffold-CM in developing new therapeutic cell-free approaches for wound healing in the field of modern medicine.

## 5. Conclusions

The combination of a biocompatible scaffold with hASC-conditioned media avoids cell therapy side-effects and presents substantial advantages for manufacturing, storage, and standardization, making it a promising biopharmaceutical. The use of a combination of growth factors, such as those present in hASC-CM, could be more advantageous compared to the addition of single growth factors and certainly better than stem cell transplantation. Even though an exhaustive characterization of the components of the conditioned culture medium, which also includes the microvesicle fraction, must be carried out, the results presented here are very encouraging and support the cell-free approach for regenerative medicine.

## Figures and Tables

**Figure 1 nanomaterials-12-01521-f001:**
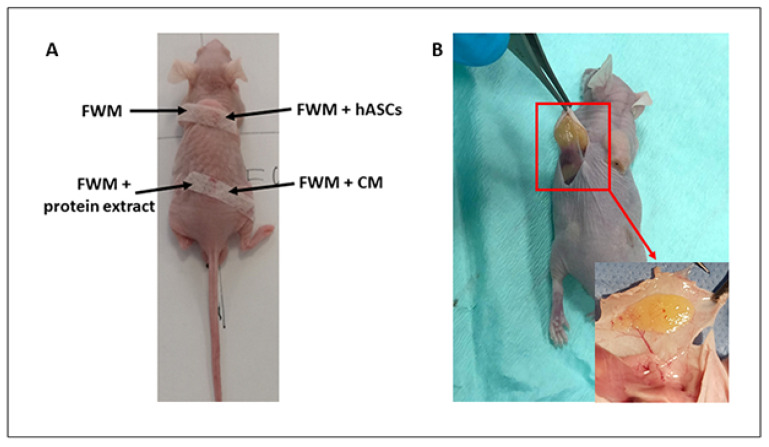
Pictures of mice’s back immediately after the grafting and the retrieval. (**A**) INTEGRA^®^ Flowable Wound Matrix grafting; (**B**) scaffold removal after 28 days with magnification on the newly formed vessels entering the scaffold.

**Figure 2 nanomaterials-12-01521-f002:**
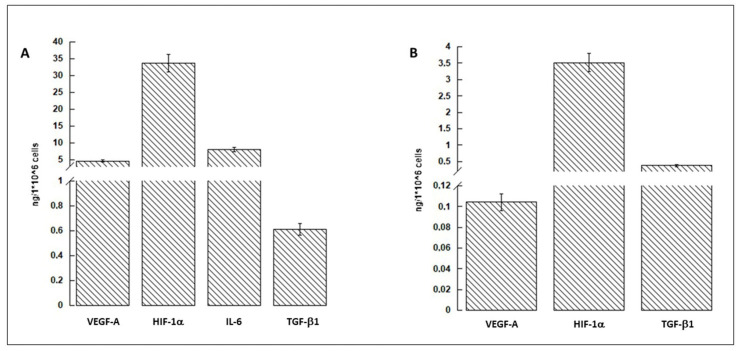
hASC-conditioned medium (**A**) and cell protein extract (**B**) characterization. VEGF-A, HIF-1α, IL-6, and TGFβ1 are evaluated. Protein amounts are given as ng/1 × 10^6^ cells. The results are expressed as mean ± S.D. n = 2.

**Figure 3 nanomaterials-12-01521-f003:**
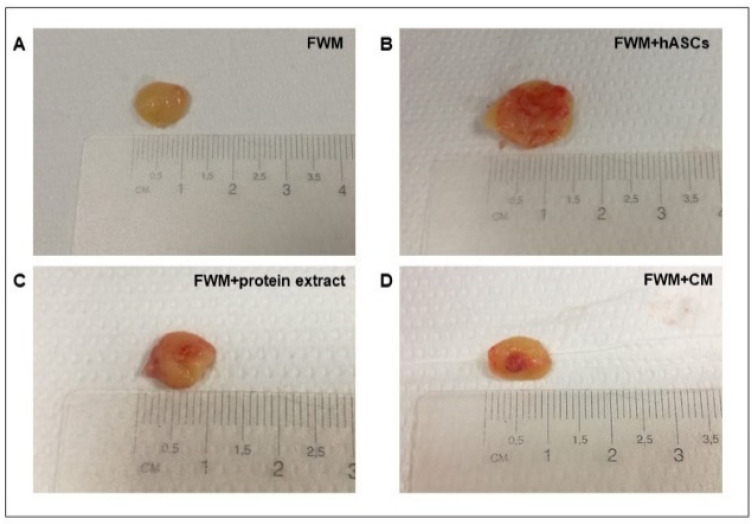
Representative images of macroscopic scaffold analysis after animal sacrifice. All the samples appeared yellow, soft on palpation and of comparable size. (**A**) FWM alone: the vascularization is almost absent. (**B**) FWM combined with hASCs. (**C**) FWM combined with cell protein extract. (**D**) FWM combined with cell-conditioned medium. In formulations (**B**–**D**), the vascularization was more evident and present also inside the scaffold.

**Figure 4 nanomaterials-12-01521-f004:**
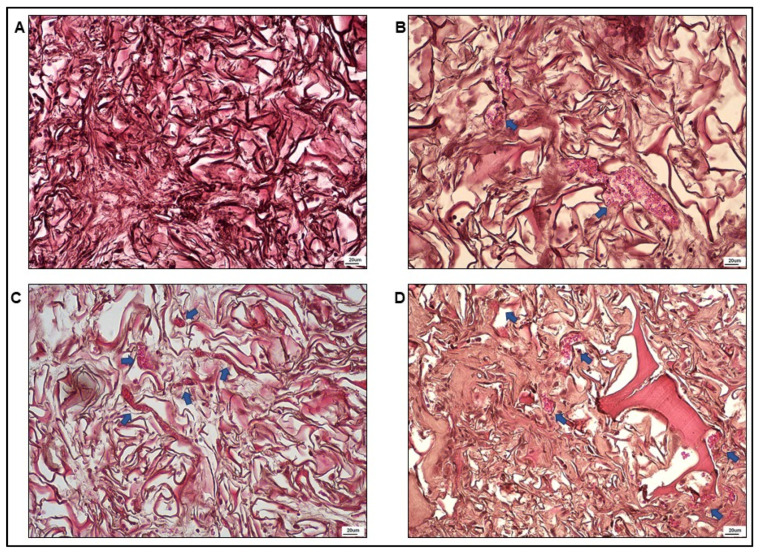
Representative microscopic images of scaffold specimens stained with hematoxylin–eosin. (**A**) FWM alone: observe the presence of collagen fibers and the absence of vessels. (**B**) FWM combined with hASCs. (**C**) FWM combined with cell protein extract. (**D**) FWM combined with cell-conditioned medium. In panels (**B**–**D**), in addition to collagen fibers, the presence of numerous cell nuclei (probably of fibroblasts) and capillaries full of erythrocytes (indicated by the blue arrows) are notable.

**Figure 5 nanomaterials-12-01521-f005:**
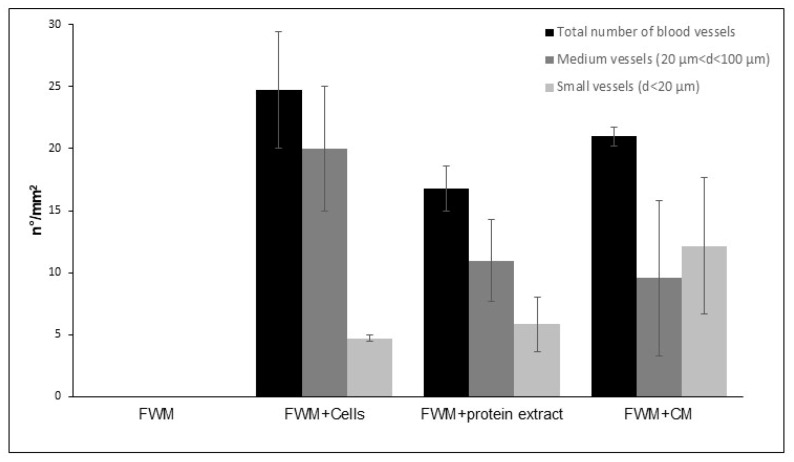
Evaluation and classification of newly formed vessels in the scaffolds represented as number of vessel/mm^2^. Although some discrepancies were appreciated among the three formulations, no statistically significant differences were observed in the number of capillaries. No blood vessels were found in the FWM alone. The results are expressed as mean ± S.D. n = 5.

**Figure 6 nanomaterials-12-01521-f006:**
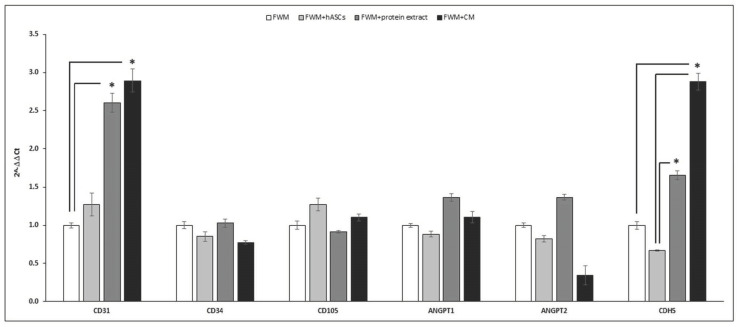
Real-time PCR of vascular growth factors in scaffold after 28 days from the grafting. mRNA expression of CD31 and CDH5 appeared statistically significantly higher in the scaffold loaded with the conditioned medium (black bar). No significant differences were observed for CD34, CD105, ANGPT1, and ANGPT2. n = 12; * *p* < 0.05.

**Table 1 nanomaterials-12-01521-t001:** Primers used in this work.

Gene Name	Sequence 5′–3′	T_m_ (°C)	Accession Number
Mm_βActin	*Fw* GCCCAGAGCAAGAGAGGTA *Rv* TAGAAGGTGTGGTGCCAGAT	65	NM_007393.5
		64.9	
Mm_GAPDH	*Fw* ACCTGCCAAGTATGATGAC *Rv* GGAGTTGCTGTTGAAGTC	64	NM_008084.3
		59.7	
Mm_CD31	*Fw* AACAGAGCCAGCAGTATGA *Rv* ATGACAACCACCGCAATG	62.6	NM_001305157.1
		62.5	
Mm_CD34	*Fw* CTGCTCCGAGTGCCATTA *Rv* CTCCTCACAACTAGATGCTTCA	63.3	NM_001111059.1
		63.7	
Mm_ANGPT1	*Fw* GGAAGATGGAAGCCTGGATT *Rv* ACTGCCTCTGACTGGTTATTG	65.1	NM_009640.4
		64.2	
Mm_ANGPT2	*Fw* CGACTACGACGACTCAGT *Rv* TCTCCACCATCTCCTTCTTC	63.7	NM_007426.4
		63.8	
Mm_CDH5	*Fw* CAGAGTCCATCGCAGAGT *Rv* AGCCAGCATCTTGAACCT	64.1	NM_009868.4
		64.4	
Mm_CD105	*Fw* CGATAGCAGCACTGGATGAC *Rv* TGGCAAGCACAAGAATGGT	64.7	NM_001146350.1
		64.5	
Hs_CD90	*Fw* CTCTACTTATCCGCCTTCACT *Rv* CGTTCTGGGAGGAGATGG	62.9	NM_006288.5
		63	
Hs_FABP4	*Fw* AAGTCAAGAGCACCATAACCT *Rv* GCATTCCACCACCAGTTTATC	63.3	NM_001442.3
		63.4	

*Fw*: forward primer; *Rv*: reverse primer.

## Data Availability

The original contributions presented in the study are included in the article, and further inquiries can be directed to the corresponding author.

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
