# Peer review of "Human Adipose-Derived Stem Cell-Conditioned Medium Promotes Vascularization of Nanostructured Scaffold Transplanted into Nude Mice"

_nanomaterials, 2022, doi:10.3390/nano12091521_

Round 1

Reviewer 1 Report

This paper was well-organized, however, I suggest a major revision before acceptance. 

1) Some abbreviations in the manuscript are unclear, such as in the abstract. 

2) Please check the whole manuscript to avoid mistyping, and the language needs to be improved. 

3) Line 63 and 64, the authors stated that some research efforts use angiogenic growth factors with scaffolds; please demonstrate more with some examples. 

4) why Integre@FWM was used in this paper? it is not clear in this paper. 

5) Even though the properties of the scaffold could be checked on the company website, I still suggest the authors summarize them, adding supporting information. 

6) The information of equipment should be given, such as the centrifuge. 

7) for 2.2 hASC isolation and culture, the hASCs were obtained from different women, so they will be two different cells because of age at 40 and 57. The authors use them together or separately?

8)The caption of figure 2 is not clear, please revise it, the indication of A B should be explained in the caption. 

9) the resolution of Figure 3 is low, please replace the photos with higher resolution. 

10) For optical microscopy, I suggest the author should provide the images with higher magnifications at the same time. 

11) How figure 5 was obtained, using some software? it is unclear in the caption and MS. 

12) The quality of figure 6 should be greatly improved. 

13) Regarding that the purpose of the scaffold was not clear in the introduction, thus the potential application was not provided in the discussion section. I suggest the author should discuss the interaction between scaffold and cells, will scaffold influence the angiogenesis? 

Author Response

This paper was well-organized, however, I suggest a major revision before acceptance. 

1) Some abbreviations in the manuscript are unclear, such as in the abstract. 

Following the Referee suggestion, in the revised paper we have reported the full name beside each abbreviation. Unfortunately, we cannot make explicit the name in the abstract due to the limited number of the words.

2) Please check the whole manuscript to avoid mistyping, and the language needs to be improved. 

Thanks for pointing that out, we have corrected the typos and then the paper has been revised by a native English speaker.

3) Line 63 and 64, the authors stated that some research efforts use angiogenic growth factors with scaffolds; please demonstrate more with some examples. 

We thank the Referee for this observation. As suggested, in the revised version we have provided more details regarding the bioactive molecules combined with the scaffold (see lines 64-65).

4) why Integre@FWM was used in this paper? it is not clear in this paper. 

The INTEGRA®FWM is a collagen scaffold commonly used for the treatment of tunneled and irregular wounds, which are often associated with excessive scar tissue formation. Being this scaffold used in a gel-like consistency, this makes it optimal for subcutaneous injection. Now we have made more explicit the reason of its use also in the text (see lines 92-97)

5) Even though the properties of the scaffold could be checked on the company website, I still suggest the authors summarize them, adding supporting information. 

As suggested, we add more information to better describe the scaffold (see lines 92-97).

6) The information of equipment should be given, such as the centrifuge. 

Considering that we expressed the centrifuge speed as “rcf”, the model is not necessary, however, in the revised version we have specified the centrifuge is an Eppendorf 5804R (see line 118-119)

7) for 2.2 hASC isolation and culture, the hASCs were obtained from different women, so they will be two different cells because of age at 40 and 57. The authors use them together or separately?

The cells obtained from the two patients have been used in two different experiments. The results obtained have been then averaged and expressed as mean±SD.

8)The caption of figure 2 is not clear, please revise it, the indication of A B should be explained in the caption. 

Thank you for pointing that out, we have modified the caption.

9) the resolution of Figure 3 is low, please replace the photos with higher resolution. 

The figures were provided as zipped file at high resolution (300dpi).

10) For optical microscopy, I suggest the author should provide the images with higher magnifications at the same time. 

The referee is right to point out the difficulty in observing the picture at this magnification and we hope that the quality of the figure, provided as zipped file at the resolution of 300dpi, is better. Unfortunately, due to the three-dimensional structure of the scaffold, acquiring images at a higher magnification did not allow us to obtain an optimal focus on the structure present inside the scaffold.

11) How figure 5 was obtained, using some software? it is unclear in the caption and MS. 

As reported at line 208, the data presented in figure 5 were obtained using “ISCapture software”.

12) The quality of figure 6 should be greatly improved. 

As previously stated, the figures are also provided as zipped file at the resolution of 300 dpi.

13) Regarding that the purpose of the scaffold was not clear in the introduction, thus the potential application was not provided in the discussion section. I suggest the author should discuss the interaction between scaffold and cells, will scaffold influence the angiogenesis? 

As reported in the introduction (lines 77-80) “in this paper, we report the results of in vivo experiments on athymic BALB/c nude mice grafted with a collagen-glycosaminoglycan biodegradable matrix (Integra® Flowable Wound Matrix) associated with human adipose derived mesenchymal stem cells (hASCs) derivatives”

Thus, we are not interested in studying the interaction between scaffold and cells.

Reviewer 2 Report

Dear authors,

I consider your work an important contribution to the field and I propose these changes in order to make the final version more easily understood.

Abstract

Lines 15 to 16 - stem cells instead staminal ones (line 357 to)

Introduction

Lines 55 to 56 - such as IL-6 , IL-10, TGF ß-1, and TSP-1

Lines 75 to 76 - Integra® Flowable Wound Matrix (FWM) instead (Integra®FWM)

Lines 74 to 84 - Here we expect to read the objectives of the work and not the main conclusions as we can read in lines 82 to 84.

Line 79 to 81 - The definition of the groups must appear in the materials and methods section.

Materials and Methods

Line 90 - Can you specify or provide a reference to the culture medium used?

Line 107 - (Body Mass Index was < 20 kg/m2).

Line 183 - Part b of Figure 2 refers to a necropsy so we cannot say "The animal treatment" for the whole of Figure 2.

Results

Line 241 - Figure 2. hASC conditioned medium (A) and cell protein extract (B) characterization.  

Lines 267 to 268 - This sentence is a repetition of a MM section sentence. "In order to take account of the formation of new vessels within the scaffold, 5 non-consecutive histological sections per sample were cut and analysed."

Discussion

Line 318 to 322 - It is not completely clear the reference to PRP in this context !

Line 354 - Cdh5 mRNA

As a minimum, the project identification code, date of approval and name of the ethics committee or institutional review board should be stated in Section ‘Institutional Review Board Statement’. In your "Institutional Review Board Statement:" I do not find "the project identification code, date of approval and name of the ethics committee or institutional review board" only general references.

Institutional Review Board Statement: All protocols were reviewed and approved by the “University of Insubria” and “Ospedale di Circolo Fondazione Macchi” Ethical Committee (24 april 2013, 423 n° 302). The European Communities Council Directive of EU/63/2010, and the Italian Ministry of Health approved the experimental study on animals.

Reviewer 3 Report

In the current manuscript, Barone et al. comparatively investigated the pro-angiogenic capability of a collagen-glycosaminoglycan biodegradable matrix (Integra®FWM) implanted in athymic BALB/c nude mice under four variants: 1) as such; 2) associated with hASCs; 3) associated with hASC-crude protein extract and 4) associated with conditioned medium (CM) derived from hADSC (FWM-CM). The results evinced the promising potential of the scaffolds loaded with hADSC-CM for regeneration of damaged tissues.

Hereinafter are shown my comments that the authors should consider:

1. Although I am not a native English speaker, I strongly recommend that a correction in terms of the written English should be applied to the entire manuscript.

2. Abstract – I suggest the authors to define the scaffolds’ composition.

3. Figures legends are too simple. Please give little more details (especially for Fig. 1-3 and 5).

4. Results, Section 3.1 – The authors are advised to comment on the lack of expression of IL-6 in protein cell extract.

5. Please expand the abbreviations at their first use within the manuscript, e.g. ANGPT1, CDH5, ROS..

6. According to my knowledge the term “staminal cells” is not equivalent to “stem cells”. Please revise throughout all manuscript.

Author Response

Comments and Suggestions for Authors

In the current manuscript, Barone et al. comparatively investigated the pro-angiogenic capability of a collagen-glycosaminoglycan biodegradable matrix (Integra®FWM) implanted in athymic BALB/c nude mice under four variants: 1) as such; 2) associated with hASCs; 3) associated with hASC-crude protein extract and 4) associated with conditioned medium (CM) derived from hADSC (FWM-CM). The results evinced the promising potential of the scaffolds loaded with hADSC-CM for regeneration of damaged tissues.

Hereinafter are shown my comments that the authors should consider:

  1. Although I am not a native English speaker, I strongly recommend that a correction in terms of the written English should be applied to the entire manuscript.

As suggested, the manuscript has been revised by a native English speaker.

  1. Abstract – I suggest the authors to define the scaffolds’ composition.

We have done (see line 18).

  1. Figures legends are too simple. Please give little more details (especially for Fig. 1-3 and 5).

As suggested, we have revised the legends of figures 1-5.

  1. Results, Section 3.1 – The authors are advised to comment on the lack of expression of IL-6 in protein cell extract.

We apologize for this lack. Now we have specified that IL-6 in protein cell extract was undetectable in our experimental conditions (see line 250-251).

  1. Please expand the abbreviations at their first use within the manuscript, e.g. ANGPT1, CDH5, ROS..

Done.

  1. According to my knowledge the term “staminal cells” is not equivalent to “stem cells”. Please revise throughout all manuscript.

We have modified as suggested.

Round 2

Reviewer 1 Report

I suggest the acceptance in its current form.